# ARTIC3D: Learning Robust Articulated 3D Shapes from Noisy Web Image Collections

**Chun-Han Yao**[1][*]    **Amit Raj**[2]    **Wei-Chih Hung**[3]    **Yuanzhen Li**[2]    **Michael Rubinstein**[2]
**Ming-Hsuan Yang**[124]    **Varun Jampani**[2]

[1]UC Merced    [2]Google Research    [3]Waymo    [4]Yonsei University

## Abstract

Estimating 3D articulated shapes like animal bodies from monocular images is inherently challenging due to the ambiguities of camera viewpoint, pose, texture, lighting, etc. We propose ARTIC3D, a self-supervised framework to reconstruct per-instance 3D shapes from a sparse image collection in-the-wild. Specifically, ARTIC3D is built upon a skeleton-based surface representation and is further guided by 2D diffusion priors from Stable Diffusion. First, we enhance the input images with occlusions/truncation via 2D diffusion to obtain cleaner mask estimates and semantic features. Second, we perform diffusion-guided 3D optimization to estimate shape and texture that are of high-fidelity and faithful to input images. We also propose a novel technique to calculate more stable image-level gradients via diffusion models compared to existing alternatives. Finally, we produce realistic animations by fine-tuning the rendered shape and texture under rigid part transformations. Extensive evaluations on multiple existing datasets as well as newly introduced noisy web image collections with occlusions and truncation demonstrate that ARTIC3D outputs are more robust to noisy images, higher quality in terms of shape and texture details, and more realistic when animated.

## 1 Introduction

Articulated 3D animal shapes are widely used in applications such as AR/VR, gaming, and content creation. However, the articulated models are usually hard to obtain as manually creating them is labor intensive and 3D scanning real animals in the lab settings is highly infeasible. In this work, we aim to automatically estimate high-quality 3D articulated animal shapes directly from sparse and noisy web image collections. This is a highly ill-posed problem due to the variations across images with diverse backgrounds, lighting, camera viewpoints, animal poses, shapes, and textures, etc. In addition, we do not assume access to any 3D shape models or per-image annotations like keypoints and camera viewpoints in our in-the-wild setting.

While several recent methods [39, 33, 38] can produce animatable 3D shapes using a skeleton-based neural surface or pre-defined mesh template, the success is largely dependent on large-scale image datasets or manually-filtered clean images for training or optimization. Moreover, the output 3D shapes and textures are usually unrealistic when viewed from novel viewpoints or pose articulations. On the other hand, recent success of generative diffusion models [25, 28, 27] shows that one can generate high-quality images for a given text prompt. DreamFusion [21] and other recent works [15, 18, 23] further demonstrate the possibility to produce 3D objects/scenes simply using 2D diffusion as multi-view supervision. In this work, we leverage the powerful 2D diffusion prior

---

[*]Work done as a student researcher at Google.

37th Conference on Neural Information Processing Systems (NeurIPS 2023).

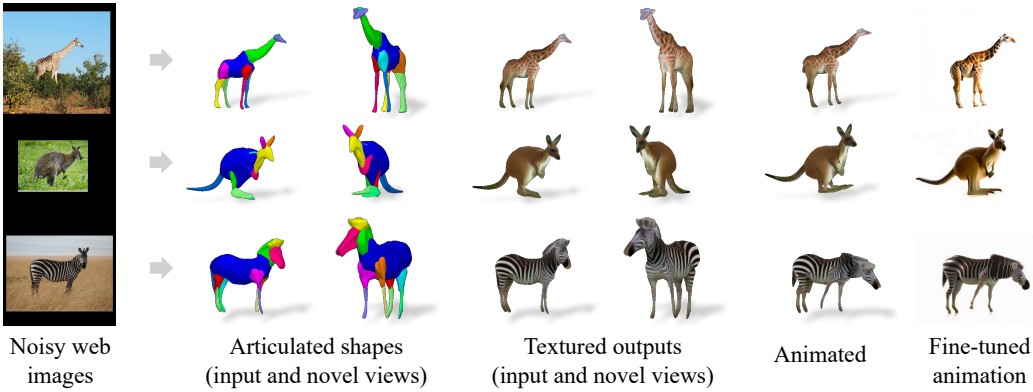

| Noisy web images | Articulated shapes (input and novel views) | Textured outputs (input and novel views) | Animated | Fine-tuned animation |

Figure 1: **Learning articulated 3D shapes from noisy web images.** We propose ARTIC3D, a diffusion-guided optimization framework to estimate the 3D shape and texture of articulated animal bodies from sparse and noisy image in-the-wild. Results show that ARTIC3D outputs are detailed, animatable, and robust to occlusions or truncation.

to learn 3D articulated shapes, aiming to reconstruct and animate 3D animals from sparse noisy online images without any 2D or 3D annotations. Intuitively, one can improve the quality of 3D reconstructions by utilizing a diffusion prior similar to the score distillation sampling (SDS) loss proposed in DreamFusion [21]. In our experiments, nonetheless, we observe that naively applying the SDS loss on 3D surface optimization leads to unstable and inefficient training, producing undesirable artifacts like noisy surfaces or ambiguous texture.

In this work, we present ARTIC3D (ARTiculated Image Collections in 3D), a diffusion-guided optimization framework to learn articulated 3D shapes from sparse noisy image collections. We use the articulated part surface and skeleton from Hi-LASSIE [38], which allows explicit part manipulation and animation. We propose a novel Decoder-based Accumulative Score Sampling (DASS) module that can effectively leverage 2D diffusion model priors from Stable Diffusion [27] for 3D optimization. In contrast to existing works that back-propagate image gradients through the latent encoder, we propose a decoder-based multi-step strategy in DASS, which we find to provide more stable gradients for 3D optimization. To deal with noisy input images, we propose an input preprocessing scheme that use diffusion model to reason about occluded or truncated regions. In addition, we also propose techniques to create realistic animations from pose articulations.

We analyze ARTIC3D on the Pascal-Part [5] and LASSIE [39] datasets. To better demonstrate the robustness to noisy images, we extend LASSIE animal dataset [39] with noisy web animal images where animals are occluded and truncated. Both qualitative and quantitative results show that ARTIC3D produces 3D shapes and textures that are detailed, faithful to input images, and robust to partial observations. Moreover, our 3D articulated representation enables explicit pose transfer and realistic animation which are not feasible for prior diffusion-guided methods with neural volumetric representations. Fig. 1 shows sample 3D reconstructions and applications from ARTIC3D. The main contributions of this work are:

- We propose a diffusion-guided optimization framework called ARTIC3D, where we reconstruct 3D articulated shapes and textures from sparse noisy online images without using any pre-defined shape templates or per-image annotations like camera viewpoint or keypoints.

- We design several strategies to efficiently incorporate 2D diffusion priors in 3D surface optimization, including input preprocessing, decoding diffused latents as image targets, pose exploration, and animation fine-tuning.

- We introduce E-LASSIE, an extended LASSIE dataset [39], by collecting and annotating noisy web images with occlusions or truncation to evaluate model robustness. Both qualitative and quantitative results show that ARTIC3D outputs have higher-fidelity compared to prior arts in terms of 3D shape details, texture, and animation.

## 2 Related Work

**Animal shape and pose estimation.** Earlier techniques on animal shape estimation used statistical body models [46, 45] that are learned either using animal figurines or a large number of annotated animal images. Some other works [35, 34, 36, 37], use video inputs to learn articulated shapes by exploiting dense correspondence information in video. However, these methods rely on optical flow correspondences between video frames, which are not available in our problem setting. Other techniques [12, 11] leverage a parametric mesh model and learn a linear blend skinning from images to obtain a posed mesh for different animal categories. MagicPony [33] learns a hybrid 3D representation of the animal instance from category specific image collections. Most related to our work are LASSIE [39] and Hi-LASSIE [38], which tackle the same problem setting of recovering 3D shapes from a sparse collection of animal images in the wild using either a manually annotated skeleton template or by discovering category-specific template from image collections. However, these approaches require carefully curated input data and fail to handle image collections with partial occlusions, truncation, or noise. By leveraging recent advances in diffusion models, we support reconstruction on a wider variety of input images.

**3D reconstruction from sparse images.** Several recent works [30, 42, 41, 32, 24, 2, 43, 3] have used implicit representations [19] to learn geometry and appearance from sparse image collections either by training in a category specific manner or assuming access to multi-view consistent sparse images during inference. However, most of these approaches demonstrate compelling results only on rigid objects. Zhang et al. [42] is another closely related work that finds a neural surface representation from sparse image collections but requires coarse camera initialization. By learning a part based mesh shape and texture, our framework naturally lends itself to modeling and animating articulated categories such as animals in the wild without any additional requirements on camera parameters.

**Diffusion prior for 3D.** Diffusion models [27, 28, 44] have recently gained popularity for generating high resolution images guided by various kinds of conditioning inputs. Diffusion models capture the distribution of real data which can be used as score function to guide 3D generation with score-distillation sampling (SDS) loss as first described in DreamFusion [21]. Several recent approaches [18, 15, 17, 29, 26, 23] leverage the SDS loss to generate 3D representations from either text or single or sparse image collections. Drawing inspiration from these lines of work, we propose a novel Decoder-based accumulative Score Sampling (DASS) that exploits the high quality images synthesized by the decoder and demonstrate improved performance over naive SDS loss.

## 3 Approach

Given 10-30 noisy web images of an animal species, ARTIC3D first preprocesses the images via 2D diffusion to obtain cleaner silhouette estimates, semantic features, and 3D skeleton initialization. We then jointly optimizes the camera viewpoint, pose articulation, part shapes and texture for each instance. Finally, we animate the 3D shapes with rigid bone transformations followed by diffusion-guided fine-tuning. Before introducing our diffusion-based strategies to improve the quality of 3D outputs, we briefly review the skeleton-based surface representation similar to [39, 38], as well as Stable Diffusion [27] that we use as diffusion prior.

### 3.1 Preliminaries

While most 3D generation methods optimizes a volumetric neural field to represent 3D rigid objects/scenes, we aim to produce 3D shapes that are articulated and animatable. To enable explicit part manipulation and realistic animation, we adopt a skeleton-based surface representation as in LASSIE [39] and Hi-LASSIE [38]. Nevertheless, unlike [39, 38] which naively sample surface texture from images, we optimize per-part texture images to obtain realistic instance textures from novel views. At a high level, [39, 38] focus on using geometry priors to learn detailed articulated shapes, whereas ARTIC3D further incorporates generative 2D diffusion priors to learn both shapes and texture in a more challenging scenario with noisy images.

**3D Skeleton.** Given a user-specified reference image in the collection, Hi-LASSIE [38] automatically discovers a 3D skeleton based on the geometric and semantic cues from DINO-ViT [4] feature clusters. The skeleton initializes a set of 3D joints and primitive part shapes, providing a good constraint of part transformation and connectivity. In our framework, we obtain cleaner feature

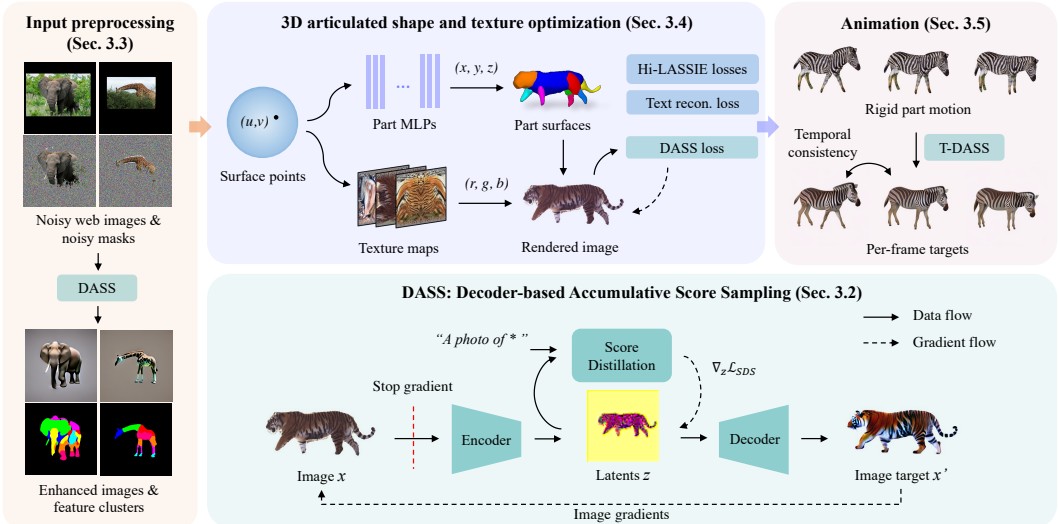

Figure 2: **ARTIC3D overview.** Given sparse web images of an animal species, ARTIC3D estimates the camera viewpoint, articulated pose, 3D part shapes, and surface texture for each instance. We propose a novel DASS module to efficiently compute image-level gradients from stable diffusion, which are applied in 1) input preprocessing, 2) shape and texture optimization, and 3) animation.

clusters by diffusing input images, then applying Hi-LASSIE as an off-the-shelf skeleton discovery method. For a fair comparison with existing works, we use the same reference image for skeleton discovery as in [38] in our experiments. Please refer to [38] for further details on skeleton discovery.

**Neural part surfaces.** Following [38], using the discovered 3D skeleton, we reconstruct a 3D part corresponding to each skeleton bone via a deformable neural surface [42]. The neural surfaces are parameterized by multi-layer perceptron networks (MLPs), mapping 3D surface points on a unit sphere to their xyz deformation. Given $m$ uniformly sampled 3D points $X \in \mathbb{R}^{3 \times m}$ on a spherical surface, we can deform the 3D shape of the $i$-th part through the part MLP as $X \mapsto \mathcal{F}_i(X)$. Then, the part surfaces are rigidly transformed by the scaling $s_i \in \mathbb{R}$, rotation $R_i \in \mathbb{R}^{3 \times 3}$, and translation $t_i \in \mathbb{R}^3$ of each skeleton part $i$. The transformed part surface points $V_i$ in the global coordinate can be written as: $V_i = s_i R_i \mathcal{F}_i(X) + t_i$ . Please refer to [38] for further details.

**Stable Diffusion architecture.** Stable Diffusion (SD) [27] is a state-of-the-art text-to-image generative model that can synthesize high-quality images given a text prompt. SD mainly consists of 3 components: An image encoder $\mathcal{E}$ that encodes a given image $x$ into a latent code $z$; a decoder network $\mathcal{D}$ that converts the latent code back to image pixels; and a U-Net denoiser $\epsilon_\phi$ that can iteratively denoise a noisy latent code. We use SD as a diffusion prior in our framework.

## 3.2 Decoder-based Accumulative Score Sampling (DASS)

To leverage the 2D diffusion prior for 3D shape learning, DreamFusion [21] proposes a score distillation sampling (SDS) loss to distill the images rendered from random views and propagate the image-level gradients to Neural Radiance Field (NeRF) parameters. To reduce the computational cost and improve training stability, recent works like Latent-NeRF [18] and Magic3D [15] perform distillation on the low-resolution latent codes in SD and back-propagate the gradients through the SD image encoder $\mathcal{E}$. Formally, let $x$ be a rendered image from 3D model and $z$ denote its latent codes from the SD image encoder $\mathcal{E}$. At each score distillation iteration, the latent codes $z$ are noised to a random time step $t$, denoted as $z_t$, and denoised by the U-Net denoiser $\epsilon_\phi$ of the diffusion model. The image-level SDS gradients can then be expressed as:

$$\nabla_x \mathcal{L}_{\text{SDS}} = w_t(\epsilon_\phi(z_t; y, t) - \epsilon)\frac{\partial z}{\partial x}, \tag{1}$$

where $y$ denotes the guiding text embedding, $\epsilon$ is the random noise added to the latent codes, and $w_t$ is a constant multiplier which depends on diffusion timestep $t$. The denoiser $\epsilon_\phi$ uses a guidance scale $w_g$ to balance the text guidance and a classifier-free guidance [8] of an unconditional model.

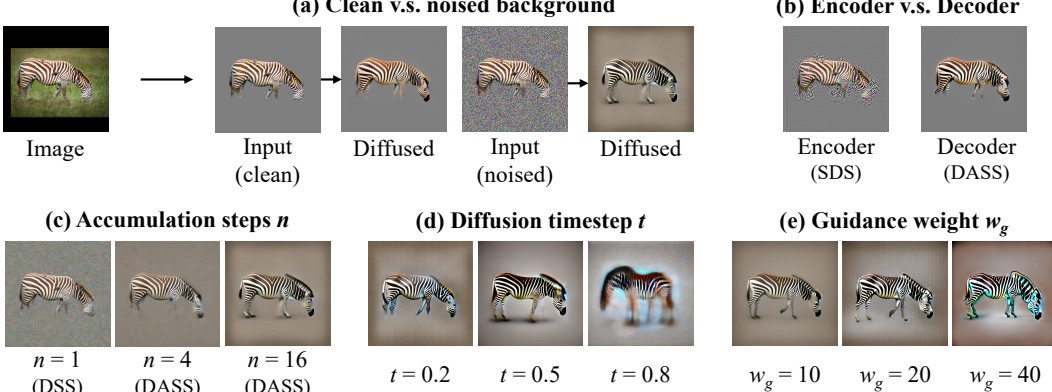

**(a) Clean v.s. noised background**

Image → Input (clean) → Diffused → Input (noised) → Diffused

**(b) Encoder v.s. Decoder**

Encoder (SDS) — Decoder (DASS)

**(c) Accumulation steps *n***

$n = 1$ (DSS) — $n = 4$ (DASS) — $n = 16$ (DASS)

**(d) Diffusion timestep *t***

$t = 0.2$ — $t = 0.5$ — $t = 0.8$

**(e) Guidance weight $w_g$**

$w_g = 10$ — $w_g = 20$ — $w_g = 40$

Figure 3: **Ablative visualizations of the DASS method.** From the example input image (top left), we show the updated image after one optimization iteration using various ways to obtain image-level gradients or parameter settings: (a) shows that noised background in the input image encourages DASS to hallucinate the missing parts; (b) compares the standard SDS (back-propagate gradients through encoder) and our DASS (decoder-based) losses; (c) justifies our accumulating latent gradient approach as it leads to cleaner decoded output; (d) indicates that small timestep mostly modifies the texture, whereas large timestep changes the geometry more (sometimes removes or creates body parts); (e) demonstrates high-contrast colors and slightly disproportioned body with higher guidance weight (diffusion prior is biased towards larger heads and frontal views). Note that (b) uses the clean input in (a) for better visualization, whereas (c),(d),(e) are obtained from the noised input.

Although this common SDS loss is effective in generating NeRFs from text, we observe that naively applying it in our framework leads to unstable and inefficient training. As shown in Fig. 3 (b), the SDS gradients back-propagated through the encoder are often quite noisy, causing undesirable artifacts on 3D shapes and texture. Moreover, it requires the extra computation and memory usage for gradient back propagation, limiting the training batch size and thus decreasing stability.

To mitigate these issues, we propose a novel Decoder-based Accumulative Score Sampling (DASS) module, an alternative to calculate pixel gradients that are cleaner and more efficient. Fig. 2 illustrates the proposed DASS module. At a high level, given an input image $x$, we obtain a denoised image $x'$ from the decoder as a reconstruction target, based on our observation that decoded outputs are generally less noisy. As shown in Fig. 2, we pass a rendered image through the encoder $\mathcal{E}$ to obtain low-resolution latent codes, update the latents for $n$ steps via score distillation, then decode the updated latents with the decoder $\mathcal{D}$ as an image target. Formally, instead of explicitly calculating the partial derivative $\partial z / \partial x$, we use $x - \mathcal{D}(z - \nabla z)$ as a proxy to $\nabla x$, where $\nabla z$ is the accumulated latent gradients over $n$ steps. This makes a linear assumption on $\mathcal{D}$ around latents $z$, which we empirically find effective to approximate the pixel gradients. The target image $x' = \mathcal{D}(z - \nabla z)$ can be directly used as an updated input (Section 3.3) or to compute a pixel-level DASS loss $\mathcal{L}_{dass} = \|(x - \mathcal{D}(z - \nabla z))\|^2$ in 3D optimization (Section 3.4). Since the DASS module only involves one forward pass of the encoder and decoder, it costs roughly half the memory consumption during training compared to the original SDS loss.

The visualizations in Fig. 3 demonstrate that DASS produces cleaner images than the original SDS loss in one training step (b), and that the accumulated gradients can effectively reduce noise and fill in the missing parts (c). Moreover, we show that adding random noise to the background pixels can facilitate the shape completion by DASS (a). We also perform ablative analyses on other diffusion parameters like noise timestep (d) and guidance weight (e) in Fig. 3. In general, ARTIC3D favors moderate accumulation steps $n \in (3, 10)$ and lower timestep $t \in (0.2, 0.5)$ since higher variance can lead to 3D results that are not faithful to the input images. Also, we use a lower guidance weight $w_g \in (10, 30)$ so that our results do not suffer from over saturation effects common in prior works due to high guidance scale in SDS loss. We apply the DASS module in three different stages in our framework: input preprocessing, shape and texture optimization, and animation fine-tuning, which we describe in the following sections.

## 3.3 Input preprocessing for noisy images

Animal bodies in real-world images often have ambiguous appearance caused by noisy texture, dim lighting, occlusions, or truncation, as shown in Fig. 4. To better deal with noisy or partial observations, we propose a novel method to enhance the image quality and complete the missing parts. Given a sparse image collection $\{I_j \in \mathbb{R}^{H \times W \times 3}\}$ ($j \in \{1, ..., N\}$ and $N$ is typically between 10-30) of an animal species, we aim to obtain accurate silhouettes estimates $\{\hat{M}_j \in \mathbb{R}^{H \times W}\}$ and clean semantic features $\{K_j \in \mathbb{R}^{h \times w \times d}\}$ for each instance. As shown in Fig. 2, we roughly estimate the foreground masks via clustering salient features extracted by a trained DINO-ViT [4] network. Then, we apply DASS to diffuse the background-masked images, resulting in animal bodies with cleaner texture and complete shapes. Formally, we obtain an updated image $I'$ by $\mathcal{D}(z - \nabla z)$, where $z = \mathcal{E}(I)$. Here, DASS serves as an image denoising and inpainting module, which can effectively generate a high-quality version of a noisy input via $n$ latent updates and a single forward pass of $\mathcal{D}$. Following the noise-and-denoise nature of diffusion models, we show in Fig. 3 (a) that manually adding Gaussian noise to the background pixels in an input image encourages DASS to hallucinate the occluded parts while mostly preserving the visible regions. Finally, we re-apply DINO-ViT feature extraction and clustering [1] on the diffused images to obtain cleaner and more complete masks as well as semantic features. Fig. 2 (left) shows sample noisy input images and the corresponding output enhanced images and feature clusters. Note that Farm3D [9] uses SD [27] to generate animal images from text for 3D training, which, however, often contain irregular shapes (*e.g.*, horses with 5 legs). On the contrary, our preprocessed images are more suitable for the sparse-image optimization framework since our goal is to reconstruct 3D shape and texture that are realistic and faithful to the input images.

## 3.4 Diffusion-guided optimization of shape and texture

Given the preprocessed images, silhouette estimates, and semantic features, we jointly optimize the camera viewpoint, pose articulation, 3D part shapes, and texture. Since we do not assume any 2D or 3D annotations, we follow Hi-LASSIE [38] and adopt an analysis-by-synthesis approach to reconstruct 3D shape and texture that are faithful to the input images. That is, we render the 3D part using a differentiable renderer [16] and compare them with the 2D images, pseudo ground-truth silhouettes, and DINO-ViT features. Fig. 2 (top) illustrates the shape and texture optimization.

**LASSIE and Hi-LASSIE losses.** Given the rendered silhouette $\tilde{M}^j$ and pseudo ground-truth $\hat{M}^j$ of instance $j$, the silhouette loss $\mathcal{L}_{sil}$ can be written as: $\mathcal{L}_{sil} = \sum_j \|\tilde{M}^j - \hat{M}^j\|^2$. LASSIE [39] and Hi-LASSIE [38] further leverage the 2D correspondence of DINO features between images of the same animal class to define a semantic consistency loss $\mathcal{L}_{sem}$. $\mathcal{L}_{sem}$ can be interpreted as the Chamfer distance between 3D surface points and 2D pixels, enforcing the aggregated 3D point features to project closer to the similar pixel features in all images. To regularize the pose articulations and part shapes, [39, 38] also apply a part rotation loss $\mathcal{L}_{rot}$, Laplacian mesh regularization $\mathcal{L}_{lap}$, and surface normal loss $\mathcal{L}_{norm}$. The part rotation loss $\mathcal{L}_{rot} = \sum_j \|R^j - \bar{R}\|^2$ limits the angle offsets from resting pose, where $R^j$ is the part rotations of instance $j$ and $\bar{R}$ denotes the part rotations of shared resting pose. $\mathcal{L}_{lap}$ and $\mathcal{L}_{norm}$ encourage smooth 3D surfaces by pulling each vertex towards the center of its neighbors and enforcing neighboring faces to have similar normals, respectively. We omit the details and refer the readers to [39, 38]. Considering that the reconstruction ($\mathcal{L}_{sil}$, $\mathcal{L}_{sem}$) and regularization ($\mathcal{L}_{rot}$, $\mathcal{L}_{lap}$, $\mathcal{L}_{norm}$) losses are generic and effective on articulated shapes, we use them in ARTIC3D along with novel texture reconstruction and DASS modules.

**Texture reconstruction.** Both [39, 38] directly sample texture from input RGB, resulting in unrealistic textures in occluded regions. To obtain more realistic textures, we also optimize a texture image $T_i$ for each part. The vertex colors $C \in \mathbb{R}^{3 \times m}$ are sampled via the pre-defined UV mapping $\mathcal{S}$ of surface points $X$. Formally, the surface color sampling of part $i$ can be expressed as $C_i = T_i(\mathcal{S}(X))$. The sampled surface texture are then symmetrized according to the symmetry plane defined in the 3D skeleton. Note that the texture images are optimized per instance since the animals in web images can have diverse texture. Similar to the $\mathcal{L}_{lap}$, we enforce the surface texture to be close to input image when rendered from the estimated input view. The texture reconstruction loss is defined as:

$$\mathcal{L}_{text} = \sum_j \|\hat{M}^j \odot (\hat{I}^j - \tilde{I}^j)\|^2, \tag{2}$$

where $\hat{I}^j$ denotes the clean input image of instance $j$ after input preprocessing and $\hat{M}^j$ denotes the corresponding animal mask; $\tilde{I}^j$ is the rendered RGB image from the estimated 3D shape and texture; and $\odot$ denotes element-wise product. The reconstruction loss is masked by the estimated foreground silhouette so that the texture optimization is only effected by the visible non-occluded animal pixels.

**Distilling 3D reconstruction.** In addition to the aforementioned losses, we propose to increase the shape and texture details by distilling 3D reconstruction. Here, we use DASS as a critic to evaluate how well a 3D reconstruction looks in its 2D renders, and calculate pixel gradients from the image target. Similar to prior diffusion-based methods [21, 18, 15], we render the 3D surfaces with random viewpoints, lighting, and background colors during training. Moreover, we design a pose exploration scheme to densify the articulation space in our sparse-image scenario. In particular, we randomly interpolate the estimated bone rotation $(R^{j_1}, R^{j_2})$ of two instances $(j_1, j_2)$, and generate a new instance with novel pose $R' = \alpha R^{j_1} + (1 - \alpha) R^{j_2}$ for rendering, where $\alpha \in (0, 1)$ is a random scalar. As such, we can better constrain the part deformation by diffusion prior and prevent irregular shape or disconnection between parts. As shown in Fig. 2, we then diffuse the latent codes of rendered images and obtain pixel gradients from the DASS module. The resulting gradients are back-propagated to update the part surface texture, deformation MLP, bone transformation, and camera viewpoints. In our experiments, we observe that the RGB gradients do not propagate well through the SoftRas [16] blending function, and we thus modify it with a layered blending approach proposed in [20].

**Optimization details.** The overall optimization objective can be expressed as the weighted sum of all the losses $\mathcal{L} = \sum_{l \in \mathfrak{L}} \alpha_l \mathcal{L}_l$, where $\mathfrak{L} = \{sil, sem, rot, lap, norm, text, dass\}$ as described above. We optimize the shared and instance-specific shapes in two stages. That is, we first update the shared part MLPs along with camera viewpoints and pose parameters. Then, we fine-tune the instance-specific part MLPs and optimize texture images for each instance. All model parameters are updated using an Adam optimizer [10]. We render the images at $512 \times 512$ resolution and at $128 \times 128$ for the part texture images. More optimization details are described in the supplemental material.

## 3.5 Animation fine-tuning

One can easily animate the resulting 3D articulated animals by gradually rotating the skeleton bones and their corresponding parts surfaces. However, the rigid part transformations often result in disconnected shapes or texture around the joints. To improve the rendered animation in 2D, one can naively use DASS frame-by-frame on a sequence of articulated shapes. However this can produce artifacts like color flickering and shape inconsistency across the frames. As a remedy, we further propose a fine-tuning step, called Temporal-DASS (T-DASS), to generate high-quality and temporally consistent 2D animations based on the ARTIC3D outputs. Given a sequence of part transformations from simple interpolation across instances or motion re-targeting, we render the 3D surfaces as video frames $\{J_k \in \mathbb{R}^{H \times W \times 3} (k \in \{1, ..., K\})\}$ and encode them into latent codes $\{z_k \in \mathbb{R}^{h \times w \times 3}\}$ through the SD encoder $\mathcal{E}$. Then, we design a reconstruction loss $\mathcal{L}_{recon}$ and temporal consistency loss $\mathcal{L}_{temp}$ to fine-tune the animation in the latent space. Similar to DASS, we obtain the reconstruction targets $\{z_k'\}$ by accumulating latent SDS gradients $\nabla z_k$ for multiple steps: $z_k' = z_k - \nabla z_k$. The reconstruction loss can then be written as: $\mathcal{L}_{recon} = \sum_t \|(z_k - z_k')\|^2$. To enforce temporal consistency, we exploit our 3D surface outputs and calculate accurate 2D correspondences across neighboring frames. Specifically, for each latent pixel in frame $z_k$, we find the closest visible 3D surfaces via mesh rasterization, then backtrack their 2D projection in frame $z_{k-1}$, forming a dense 2D flow field $F_k \in \mathbb{R}^{h \times w \times 2}$. Intuitively, the corresponding pixels should have similar latent codes. Hence, we use $F_k$ to perform temporal warpping on the latent codes $z_{k-1}$, denoted as: $\text{warp}(z_{k-1}, F_k)$, and define $\mathcal{L}_{temp}$ as:

$$\mathcal{L}_{temp} = \sum_{k=2}^{K} \|(z_k - \text{warp}(z_{k-1}, F_k)\|^2. \tag{3}$$

We fine-tune the latent codes $\{z_k\}$ with $\mathcal{L}_{recon}$ and $\mathcal{L}_{temp}$, where $\{F_k\}$ are pre-computed and $\{z_k'\}$ are updated in each iteration. Finally, we can simply obtain the RGB video frames by passing the optimized latent codes through the SD decoder $\{\mathcal{D}(z_k)\}$. The proposed $\mathcal{L}_{recon}$ encourages better shape and texture details in each frame, and $\mathcal{L}_{temp}$ can effectively regularize latent updates temporally. Note that T-DASS optimizes the latent codes and takes temporal consistency into account, which is different from DASS which operates on each image individually.

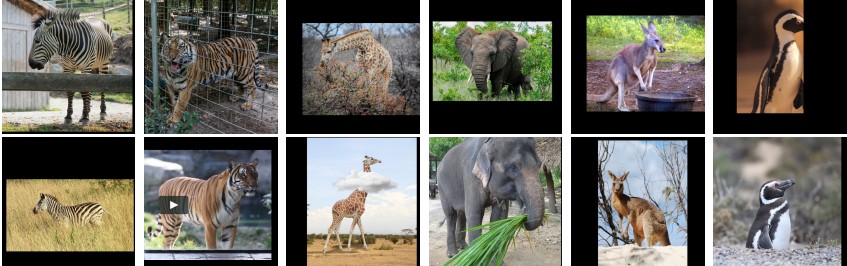

Figure 4: **E-LASSIE samples.** We extend LASSIE [39] image sets with 15 occluded or truncated images per animal class and annotate the 2D keypoints for evaluation. These noisy images pose great challenges to sparse-image optimization since the per-instance 3D shapes can easily overfit to the visible parts and ignore the rest.

## 4 Experiments

**Datasets.** Following [39, 38], we evaluate ARTIC3D on the Pascal-Part [5] and LASSIE [39] images. From Pascal-Part, we obtain images of horse, cow, and sheep, as well as their 2D keypoints automatically computed using the ground-truth 2D part masks. The LASSIE dataset includes web images of other animal species (zebra, tiger, giraffe, elephant, kangaroo, and penguin) and 2D keypoint annotations. Each image collection contains roughly 30 images of different instances with diverse appearances, which are manually filtered so that the animal bodies are fully visible in the images. To evaluate the model robustness in a more practical setting, we extend the LASSIE image sets with several noisy images where the animals are occluded or truncated. In particular, we collect 15 additional web images (CC-licensed) per class and annotate the 2D keypoints for evaluation. We call the extended image sets E-LASSIE and show some examples in Fig. 4. For the experiments on E-LASSIE, we optimize and evaluate on all the 45 images in each set.

**Baselines.** We mainly compare ARTIC3D with LASSIE [39] and Hi-LASSIE [38] as we deal with the same problem setting, namely sparse image optimization for articulated animal shapes. For reference, we also compare the results with several learning-based methods like A-CSM [11], MagicPony [15], and Farm3D [9]. Note that these approaches are not directly comparable to ARTIC3D since they train a feedforward network on large-scale image sets (not available in our scenario). Although related, some other recent works on 3D surface reconstruction either cannot handle articulations [12, 14, 7, 31, 40] or require different inputs [13, 34, 36]. As a stronger baseline, we implement Hi-LASSIE+, incorporating the standard SDS loss as in [27, 15, 18] (back-propagate latent gradients through encoder) during Hi-LASSIE [38] optimization for shape and texture.

**Evaluation metrics.** Considering the lack of ground-truth 3D annotations in our datasets, we follow a common practice [45, 11, 39, 38] to use keypoint transfer accuracy as a quantitative metric to evaluate 3D reconstruction. For each pair of images, we map the annotated 2D keypoints on source image onto the canonical 3D surfaces, re-project them to the target image via the estimated camera, pose, and shape, and compare the transferred keypoints with target annotations. To further evaluate the quality of textured outputs, we compute CLIP [22] features of the 3D output renders under densely sampled viewpoints, and calculate the feature similarity against text prompt as well as input images. While most prior arts on 3D shape generation [21, 23] only evaluate the image-text similarity, we also evaluate the image-image similarity since our outputs should be faithful to both the category-level textual description as well as instance-specific input images. We use a text prompt: "A photo of *" for each animal class "*" in our experiments. A CLIP ViT-B/32 model is used to compute the average feature similarity over 36 uniformly sampled azimuth renders at a fixed elevation of 30 degrees. We show the main results here and more quantitative and qualitative comparisons in the supplemental material, including animation videos, user study, and more detailed ablation study.

**Qualitative results.** Fig. 1 shows some sample outputs of ARTIC3D. In Fig. 5, we compare the visual results of Hi-LASSIE, Hi-LASSIE+, and ARTIC3D on the E-LASSIE images. Both Hi-LASSIE and Hi-LASSIE+ produce irregular pose and shape for the invisible parts. Regarding surface texture, Hi-LASSIE reconstructs faithful texture from the input view but noisy in novel views, since it naively samples vertex colors from the input images. The output texture of Hi-LASSIE+ is generally less

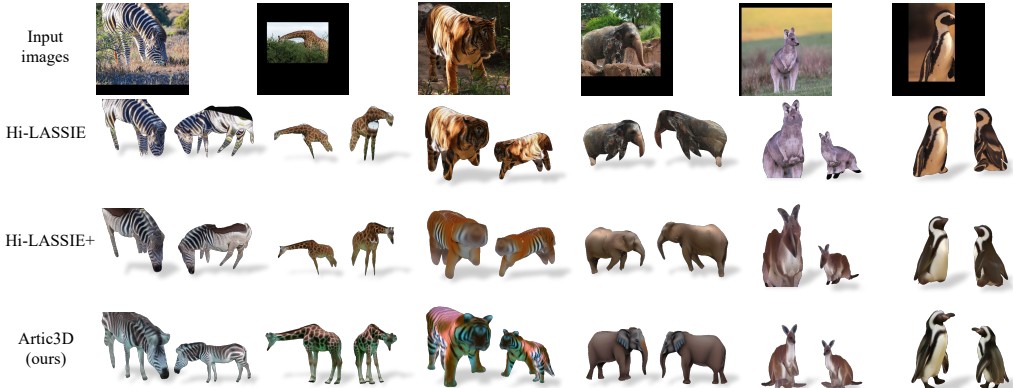

Figure 5: **Visual comparison of ARTIC3D and other baselines.** For each input image, we show the 3D textured outputs from input (upper) and novel (lower) views. The results demonstrate that ARTIC3D is more robust to noisy images with occlusions or truncation, producing 3D shape and texture that are detailed and faithful to the input images.

noisy thanks to the SDS loss. By comparison, ARTIC3D accurately estimates the camera viewpoint, pose, shape, and texture even with the presence of occlusions or truncation. Note that there exists a tradeoff between realism and faithfulness to input images due to the highly ill-posed nature of our problem setting, especially for the unseen/occluded surface. For instance, whether a colored or black-and-white output is preferred given a grey-scale/poorly illuminated image, or whether we preserve the noisy texture (small occlusions like dirt, water splash, or shadow caused by rough surface / objects) in the original image. Since the dense surface visibility is hard to obtain, we optimize the shape and texture that are slightly biased towards realism (detailed and clean texture that resembles the animal class) in this paper. In our ablative analysis of the DASS module (Fig. 3), we show that we can control the realism-faithfulness tradeoff by tuning the diffusion timestep $t$ and guidance weight $w_g$. Specifically, larger $t$ and $w_g$ allows DASS to hallucinate shape and texture that are not present in the original image. In addition, one can enforce a more faithful reconstruction by setting a higher weight of the texture reconstruction loss $\alpha_{text}$. More qualitative comparisons against other baselines can be found in the supplemental material.

**Quantitative comparisons.** We show comparisons of the keypoint transfer accuracy (PCK) in Tables 1. On both LASSIE and E-LASSIE image sets, Hi-LASSIE+ produces a marginal PCK gain from Hi-LASSIE [38] by naively applying the SDS loss. ARTIC3D, on the other hand, achieves consistently higher PCK than the baselines, especially on the noisy E-LASSIE images. The results demonstrate that our diffusion-guided strategies can effectively learn more detailed, accurate, and robust 3D shapes. The Pascal-Part results in Tab 2 further show that ARTIC3D performs favorably against the state-of-the-art optimization-based methods and are comparable to learning-based approaches. In Table 3, we show the CLIP similarity comparisons on the E-LASSIE images, which indicate that our textured outputs are more faithful to both the input images (instance-level) and text prompt (class-level) for most animal classes.

**Animation and texture transfer.** In Fig. 6, we compare the animations before and after our fine-tuning step via T-DASS. While the skeleton-based representation allows easy animation via rigid part transformation, the output part shapes and texture are often disconnected and irregular around the joints. The results show that T-DASS can effectively produce high-quality animations that are detailed in shape and texture and temporally consistent between frames. In addition to animation, our 3D part surfaces also enables convenient controllable syntheses like texture transfer and pose transfer between different instance or animal classes. Several examples of texture transfer are shown in Fig. 7. More visual results with video results of these applications are shown in the supplemental material.

**Limitations.** ARTIC3D relies on the 3D skeleton discovered by Hi-LASSIE [38] to initialize the parts. If the animal bodies are occluded or truncated in most images, the skeleton initialization tends to be inaccurate, and thus limiting ARTIC3D's ability to form realistic parts. Although our input preprocessing method can mitigate this issue to some extent, fluffy animals (*e.g.* sheep) with

Table 1: **Keypoint transfer evaluations on the LASSIE [39] and E-LASSIE image sets.** We report the average PCK@0.05 (↑) on all pairs of images. ARTIC3D performs favorably against the optimization-based prior arts on all animal classes. The larger performance gap in the E-LASSIE demonstrates that ARTIC3D is robust to noisy images.

| Method | Image set | Elephant | Giraffe | Kangaroo | Penguin | Tiger | Zebra |
|---|---|---|---|---|---|---|---|
| LASSIE [39] | LASSIE | 40.3 | 60.5 | 31.5 | 40.6 | 62.4 | 63.3 |
| Hi-LASSIE [38] | LASSIE | 42.7 | 61.6 | 35.0 | 44.4 | 63.1 | 64.2 |
| Hi-LASSIE+ | LASSIE | 43.3 | 61.5 | 35.5 | 44.6 | 63.4 | 64.0 |
| ARTIC3D | LASSIE | **44.1** | **61.9** | **36.7** | **45.3** | **64.0** | **64.8** |
| Hi-LASSIE [38] | E-LASSIE | 37.6 | 54.3 | 31.9 | 41.7 | 57.4 | 60.1 |
| Hi-LASSIE+ | E-LASSIE | 38.3 | 54.8 | 32.8 | 41.8 | 57.7 | 61.3 |
| ARTIC3D | E-LASSIE | **39.8** | **58.0** | **35.3** | **43.8** | **59.3** | **63.0** |

Table 2: **Keypoint transfer results on Pascal-Part [6].** We report the mean PCK@0.1 (↑) on all pairs of images. * indicates learning-based models which are trained on a large-scale image set.

| Method | Horse | Cow | Sheep |
|---|---|---|---|
| UMR* [14] | 24.4 | - | - |
| A-CSM* [11] | 32.9 | 26.3 | 28.6 |
| MagicPony* [33] | **42.9** | **42.5** | 26.2 |
| Farm3D* [9] | 42.5 | 40.2 | **32.8** |
| LASSIE [39] | 42.2 | 37.5 | 27.5 |
| Hi-LASSIE [38] | 43.7 | 42.1 | 29.9 |
| Hi-LASSIE+ | 43.3 | 42.3 | 30.5 |
| ARTIC3D | **44.4** | **43.0** | 31.9 |

Table 3: **CLIP similarity (↑) evaluations on the E-LASSIE images.** For each animal class, we calculate cosine similarities $s1/s2$, where $s1$ is the image-image similarity (against masked input image) and $s2$ is the image-text similarity (against text prompt).

| Method | Elephant | Giraffe | Kangaroo | Penguin | Tiger | Zebra |
|---|---|---|---|---|---|---|
| Hi-LASSIE [38] | 80.0 / 26.3 | 85.2 / 29.6 | 77.4 / 25.6 | **85.8** / 30.8 | 79.7 / 25.6 | 83.8 / 27.4 |
| Hi-LASSIE+ | 79.0 / 27.7 | 84.7 / 30.2 | 78.3 / 29.1 | 82.9 / 32.3 | 75.3 / 25.3 | 81.9 / 27.6 |
| ARTIC3D | **82.6 / 28.4** | **85.3 / 30.7** | **81.6 / 29.9** | 85.5 / **33.1** | **80.0 / 27.8** | **84.1 / 29.4** |

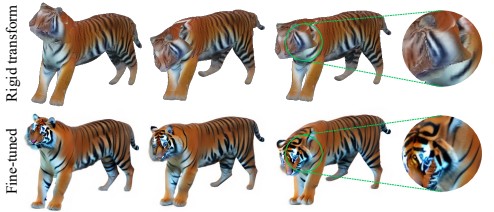 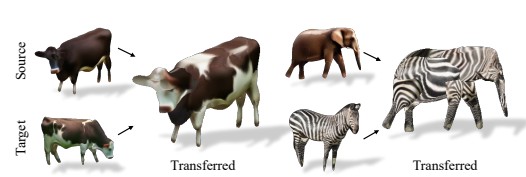

Figure 6: **Animation fine-tuning.** Compared to the original animated outputs via rigid transformation (top), our animation fine-tuning (bottom) effectively improves the shape and texture details, especially around animal joints.

Figure 7: **Texture transfer.** Our part surface representation enables applications like pose or texture transfer. Given a source shape and target texture, we show the transferred texture between instances (left) and animal species (right).

ambiguous skeletal configuration can still pose challenges in skeleton discovery. In addition, the front-facing bias in diffusion models sometimes lead to unrealistic texture like multiple faces, which also affects our reconstruction quality. See the supplemental material for failure cases.

## 5 Conclusion

We propose ARTIC3D, a diffusion-guided framework to reconstruct 3D articulated shapes and texture from sparse and noisy web images. Specifically, we design a novel DASS module to efficiently calculate pixel gradients from score distillation for 3D surface optimization. We use the DASS module and its variants in the input preprocessing of noisy images; shape and texture optimization; as well as the animation fine-tuning stages. In addition to the existing Pascal-Part and LASSIE datasets, we also evaluate on the self-collected E-LASSIE images with occlusions and truncation. Results on these datasets demonstrate that ARTIC3D produces more robust, detailed, and realistic reconstructions against prior arts. As our future work, we hope to extend ARTIC3D to more general articulated objects.

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
