# OpenReview forum: "ARTIC3D: Learning Robust Articulated 3D Shapes from Noisy Web Image Collections"
_NeurIPS.cc/2023/Conference — NeurIPS 2023 poster_

### Official Review · Reviewer_atAr · 2023-07-04

**Soundness:** 3 good
**Presentation:** 2 fair
**Contribution:** 2 fair
**Rating:** 5
**Confidence:** 4

**Summary:**

ARTIC3D is a self-supervised framework that reconstructs 3D articulated shapes and textures of animals from sparse and noisy online images. It uses a skeleton-based surface representation and 2D diffusion priors to enhance the input images and guide the 3D optimization. It also enables realistic animation by fine-tuning the rendered shape and texture under rigid part transformations. ARTIC3D outperforms prior methods in terms of shape and texture fidelity, robustness to occlusions and truncation, and pose transferability. The authors also introduce E-LASSIE, an extended dataset with noisy web images, to evaluate model robustness.

**Strengths:**

* ARTIC3D is a self-supervised framework that can reconstruct 3D articulated shapes and textures of animals from sparse and noisy online images, without relying on any pre-defined shape templates or per-image annotations. This makes it scalable and adaptable to different animal species and poses.
* The method leverages 2D diffusion to enhance the input images by removing occlusions and truncation, and to extract semantic features and 3D skeleton initialization. This improves the quality and robustness of the 3D outputs, as well as the efficiency and stability of the optimization process.
* Usage of diffusion-guided 3D optimization to estimate shape and texture that are faithful to the input images and consistent across different viewpoints and poses. It also introduces a novel technique to calculate more stable image-level gradients via diffusion models, which enhances the convergence and robustness of the optimization.
* ARTIC3D produces realistic animations by fine-tuning the rendered shape and texture under rigid part transformations, which preserves the articulation and details of the 3D shapes. It also enables explicit pose transfer and animation, which are not feasible for prior diffusion-guided methods with neural volumetric representations.

**Weaknesses:**

* The manuscript is not easy to follow for anybody who is unfamiliar with LASSIE and HI-LASSIE. This work is a step forward from Hi-LASSIE using Stable Diffusion to avoid several pitfalls with respect to optimization and image pre-processing.

* ARTIC3D depends on the 3D skeleton initialization from Hi-LASSIE [38], which may be inaccurate for occluded or truncated animal bodies, resulting in unrealistic part shapes. Also, it struggles with fluffy animals with ambiguous skeletal configuration, such as sheep, which pose challenges in skeleton discovery and shape reconstruction.

* The reconstruction results are not significantly better than Hi-LASSIE in most case 1% better, however it is not clear from the manuscript whether this is statistically significant or not.

**Questions:**

* The usage of SD improves texture quality on a perceptual level. However, as seen in many figures (ex 3, 4 in supp, 3 main) it mainly hallucinates the texture to appear realistic however further apart from the image used as reference. For example, the reconstructed elephant looks nothing like the image, similarly the kangaroo, tiger, zebra. The work is presented as reconstruction work and as such diverging from the images can't be considered a proper reconstruction. Have you examined a way to mitigate this texture drift?

* The user studies on animation are flawed. A 55% preference from 100 users means that ARTIC3D'a animations are only slightly better than rigid transform which is close enough to random choice. Could you explain in detail what sample size was used for the user study? Whether the studies were cherry picked prior to the user study? I find the user study explanation to lack a lot of key details.

**Limitations:**

Yes

---

> ### Author Rebuttal · Authors · 2023-08-09
>
> ---
> **”More details of LASSIE and Hi-LASSIE are needed”**
>
> We thank the reviewer for the feedback and will add more details to the preliminaries section (3.1) in the manuscript.
>
>
> ---
> **”3D skeleton from Hi-LASSIE may not work for occluded/truncated images”**
>
> Our 3D skeleton initialization is performed on the reference image after input preprocessing with DASS, which can effectively complete the occluded or truncated animal bodies. That is, our skeleton is more robust and accurate compared to Hi-LASSIE in the partial-body cases. Fluffy animals are a common challenge in our ill-posed problem setting. However, we are able to produce reasonable reconstruction of sheep as shown in supplemental Figure 8, which is a considerable improvement from prior works.
>
>
> ---
> **Reconstruction performance gain**
>
> As shown in Table 1 of the main paper, ARTIC3D achieves 0.6-1.7% PCK gain on the clean LASSIE images and 1.9-3.7% PCK gain on the noisy E-LASSIE images compared to Hi-LASSIE. It demonstrates a consistent advantage on ARTIC3D over Hi-LASSIE on all animal classes, especially in the occlusion/truncation cases. Moreover, the CLIP similarity evaluation in Table 3 also shows consistently favorable textured reconstructions by ARTIC3D. For more detailed discussion, please see “Contribution beyond LASSIE / Hi-LASSIE” in the General Response above.
>
>
> ---
> **Unfaithful texture from Stable Diffusion**
>
> Please see “Unfaithful texture from Stable Diffusion” in the General Response above.
>
>
> ---
> **Details of user study**
>
> In our user study, we randomly select 3 examples per animal class from the E-LASSIE and Pascal-Part datasets. More details of the user study will be added to the manuscript. As shown in the video and discussed in supplemental Sec. 2.3, rigid transformation produces static texture during motion and undesirable gaps around joints, and per-frame DASS outputs contain sharp details but are flickering and temporally inconsistent. We propose T-DASS as an alternative to find a better tradeoff between high-fidelity details and temporal smoothness. Despite the small gap, the user study still shows an overall preference of T-DASS considering both realism and temporal consistency. Please also note that all three methods (rigid transformation,  DASS, and T-DASS) are our contributions, which enable animation from a **single image without any annotations** and that no prior method can achieve this.

---

> ### Author Response · Authors · 2023-08-14
> **Please let us know whether you have additional questions after reading our response**
>
> We appreciate your reviews and comments. We hope our responses address your concerns. Please let us know if you have further questions after reading our rebuttal.
>
> We hope to address all the potential issues during the discussion period.

---

> > ### Author Response · Authors · 2023-08-17
> > **Please let us know whether all questions have been addressed**
> >
> > Dear Reviewer,
> >
> > As we are approaching the midpoint of the discussion period, we would like to confirm whether we have successfully addressed the raised concerns in your review. Should any lingering issues require further attention, please let us know as early as possible so we can answer them soon.
> >
> > We appreciate your time and effort in enhancing the quality of our manuscript.
> >
> > Thank you

---

### Official Review · Reviewer_GaMw · 2023-07-05

**Soundness:** 3 good
**Presentation:** 3 good
**Contribution:** 2 fair
**Rating:** 5
**Confidence:** 4

**Summary:**

This paper proposes a method to reconstruct the shape and texture of articulated objects from noisy web image collections. To achieve this, ARTIC3D proposed a diffusion-based 2D image enhancement module DASS, and then reconstructed the shape and texture maps using Hi-LASSIE. Moreover, to increase the animation results, ARTIC3D introduced a T-DASS module for animation fine-tuning. Experiments on the E-LASSIE dataset show that this method can produce high-fidelity animation results from noisy inputs.

**Strengths:**

--ARTIC3D can directly reconstruct the shapes and texture maps of articulated objects from noisy web image collections, greetly increase the robotness of Hi-LASSIE.
--The proposed DASS and T-DASS modules are novel, intuitive and effective.
--The paper is well-writen and easy to follow.

**Weaknesses:**

--The texture map obtained by ARTIC3D is not good enough. To solve this problem, ARTIC3D relies on the diffusion-based DASS module and the animation fine-tuning module to achieve high-fidelity animation and novel view rendering results. However, this may lead to 3D inconsistency. How does this method solve this problem?
--Although enhanced by T-DASS, the animation results are still blurry and the texture moves over time. Also, the T-DASS module seems to make the results blurrier than directly using DASS module.
--The reconstructed texture and shape are not faithful to the input image in some cases. The DASS module change the shape and appearance of the input images. Moreover, the images generated by this module may lose their 3D consistency.

**Questions:**

--ARTIC3D optimized the textured images per instance(L220-211). However, the appearance of a single animal may also vary with the lighting conditions. How does this method deal with this problem?
--ARTIC3D ustilizes the T-DASS module to enhance animation. It would be great if the paper can also include some discussions on rendering speed and other relevant computational costs.
--The T-DASS module may lead to 3D inconsistency. How does this method solve this problem?

**Limitations:**

--The shapes and textures generated by this method is inaccurate and blurry, and are not faithful to the input images in some cases.
--The DASS module may lead to 3D inconsistency.

---

> ### Author Rebuttal · Authors · 2023-08-09
>
> ---
> **3D inconsistency and unfaithful texture from Stable Diffusion**
>
> Most diffusion-based 3D generation methods like DreamFusion [21] only rely on the 2D diffusion prior to produce 3D shapes and texture, and thus they are prone to inconsistent outputs from different views (e.g. multiple faces on one animal body). ARTIC3D, on the other hand, combines the 2D diffusion prior, semantic correspondence, and 3D geometric priors (silhouettes from multiple views/poses, 3D part surface and pose regularizations, etc), which largely mitigates the 3D inconsistency issue. For more discussion on texture reconstruction, please see “Unfaithful texture from Stable Diffusion” in the General Response above.
>
>
> ---
> **Blurry animation results with T-DASS**
>
> As discussed in supplemental Sec. 2.3, rigid transformation produces static texture during motion and undesirable gaps around joints, and per-frame DASS outputs contain sharp details but are flickering and temporally inconsistent. We propose T-DASS as an alternative to find a better tradeoff between high-fidelity details and temporal smoothness. Although the tradeoff might be suboptimal, we argue that T-DASS makes a good contribution considering that the animation is created from a **single image without any annotations** and that no prior method can achieve this.
>
>
> ---
> **Lighting conditions**
>
> Considering that our problem setting is highly ill-posed (sparse, unannotated, and uncorrelated images), we do not explicitly model lighting and thus the texture image incorporates both albedo and lighting information of the input image. Note that our diffusion-based DASS module can recover natural texture from poorly illuminated or shadowed images.
>
>
> ---
> **Rendering speed and computational costs**
>
> To render 30 frames at 512x512 resolution with rigid part transformation, it takes roughly 2-3 minutes on a single GTX 1080 GPU.	 Fine-tuning the animation frames with the DASS module for 300 iterations increases the total rendering time to 8 minutes. With the T-DASS module, the total fine-tuning + rendering time is roughly 10 minutes. Since the 2D flow fields and temporal consistency loss are computed on the low-resolution (64x64) feature maps, the T-DASS module marginally increases the rendering time by 2 minutes per 30 frames to enforce temporal consistency.

---

> ### Author Response · Authors · 2023-08-14
> **Please let us know whether you have additional questions after reading our response**
>
> We appreciate your reviews and comments. We hope our responses address your concerns. Please let us know if you have further questions after reading our rebuttal.
>
> We hope to address all the potential issues during the discussion period.

---

> > ### Author Response · Authors · 2023-08-17
> > **Please let us know whether all questions have been addressed**
> >
> > Dear Reviewer,
> >
> > As we are approaching the midpoint of the discussion period, we would like to confirm whether we have successfully addressed the raised concerns in your review. Should any lingering issues require further attention, please let us know as early as possible so we can answer them soon.
> >
> > We appreciate your time and effort in enhancing the quality of our manuscript.
> >
> > Thank you

---

### Official Review · Reviewer_V4Hg · 2023-07-06

**Soundness:** 3 good
**Presentation:** 3 good
**Contribution:** 3 good
**Rating:** 6
**Confidence:** 5

**Summary:**

This paper proposes a new framework, named ARTIC3D, to address the task of 3D reconstruction of articulated shapes and texture from noisy and few images. It is based on pre-trained diffusion models. Specifically, the authors use a novel decoder-based accumulative score sampling (DASS) to replace score distillation sampling (SDS). This has been used in many other frameworks to calculate pixel gradients in order to use diffusion priors more efficiently. Besides, they also extend the LASSIE dataset with more annotated data, called E-LASSIE, which could be useful for future works , especially for robustness of 3D reconstruction models.

**Strengths:**

(1) Good writing.

(2) Plentiful experiments to prove the effectiveness of their framework and necessity of each module.

(3) Novelty of the method (DASS) to better implement 2D diffusion priors in 3D reconstruction tasks, especially when big and clean datasets are not available. Detailed techniques are elaborated in the method section, including how to use them in preprocessing noisy input image, shape and texture optimization, and animation fine-tuning.

(4) Extension of the LASSIE dataset with more annotated images to a new dataset, E-LASSIE, which could be useful for future research in this area, especially for evaluating the 3D reconstruction model robustness.

**Weaknesses:**

(1) This framework is highly based on LASSIE, both the method and the necessary input. Specifically, this framework needs the 3D skeleton from a pre-trained LASSIE model. And the framework also shares many parts with LASSIE's. But as mentioned in the Strengths part, the authors propose a new method to better incorporate diffusion models in 3D reconstruction and they propose a new dataset.

(2) Though the title claims "learning from noisy web image collections", the noise actually only involves truncation and occlusions. There are also many other types of noise that have not been explored, including illumination variations, too small instances, multiple instances, etc., which are common in web images. They also uses DINO-VIT to do the foreground-background segmentation, which is acceptable but tricky, since the noisy background is also an important and common difficulty when dealing with web images.

**Questions:**

n/a

**Limitations:**

Limitations have already been elaborated clearly in their paper.

---

> ### Author Rebuttal · Authors · 2023-08-09
>
> ---
> **Contribution beyond LASSIE / Hi-LASSIE**
>
> Please see “Contribution beyond LASSIE / Hi-LASSIE” in the General Response above.
>
>
> ---
> **”There are many other types of noises that are not explored”**
>
> We thank the reviewer for the feedback and will revise the wording in the manuscript. Please note that ARTIC3D can deal with general noise beyond occlusion/truncation to some extent as long as the DINO-ViT feature clusters are robust. For instance, several images in the E-LASSIE dataset are either greyscale, poorly illuminated, or contain small instances and noisy backgrounds, as shown in the examples through this [anonymous link](https://www.dropbox.com/scl/fi/ykuubnkgn1j5emo6d4tl9/noisy_images.png?rlkey=mfzegdj9az9t5q0wl34ql27xv&dl=0). Although not within the scope of this paper, automatic filtering of web images is an important next step towards real-world application.

---

> ### Author Response · Authors · 2023-08-14
> **Please let us know whether you have additional questions after reading our response**
>
> We appreciate your reviews and comments. We hope our responses address your concerns. Please let us know if you have further questions after reading our rebuttal.
>
> We hope to address all the potential issues during the discussion period.

---

> > ### Author Response · Authors · 2023-08-17
> > **Please let us know whether all questions have been addressed**
> >
> > Dear Reviewer,
> >
> > As we are approaching the midpoint of the discussion period, we would like to confirm whether we have successfully addressed the raised concerns in your review. Should any lingering issues require further attention, please let us know as early as possible so we can answer them soon.
> >
> > We appreciate your time and effort in enhancing the quality of our manuscript.
> >
> > Thank you

---

> > > ### Comment · Reviewer_V4Hg · 2023-08-18
> > >
> > > I thank the authors for the clarifications. I will keep my initial rating.

---

### Official Review · Reviewer_MsPL · 2023-07-07

**Soundness:** 3 good
**Presentation:** 3 good
**Contribution:** 2 fair
**Rating:** 6
**Confidence:** 4

**Summary:**

This paper introduces an articulated 3D shape reconstruction method from noisy web images with the help of diffusion models. The authors use a diffusion method to enhance the noisy input images to get clean reference 2D images and masks. Then, skeleton-based surface representations are optimized from the reference images. Then, fine-tuning improves the animations of the reconstructed shapes.

**Strengths:**

1. The overall narrative of the paper is sound and readable.
2. The authors propose to produce clean input reference images from noisy web images using diffusion models as a preprocessing step.
3. The authors propose Decoder-based Accumulative Score Sampling  (DASS) to improve efficiency and reduce artifacts.
4. The authors designed a fine-tuning step to allow better animation of the reconstructed objects.

**Weaknesses:**

1. The reconstruction part heavily relies on previous works like LASSIE[39] and Hi-LASSIE[38], it seems that the authors did not contribute very much to the core algorithm in this reconstruction task. Diffusion models helped improve the results, but most of the contribution is in the data cleaning and preprocessing.
2. Diffusion models help in image preprocessing, but Stable Diffusion would also produce results from its knowledge based on the text prompt. Therefore we can observe that some textures of the reconstructed objects are different from the reference input images, even for the unoccluded parts.
3. Figure 2 is not clear enough to show the whole workflow of the proposed methods, it is hard to relate the DASS module with the shape and texture optimization.
4. Lacking ablation studies, it is unclear whether the Distilling 3D reconstruction is working in section 3.4, without the Distilling 3D reconstruction part, the reconstruction technique would be mostly relied on LASSIE[39] and Hi-LASSIE[38] as I mentioned above.

**Questions:**

1. How does the diffusion-guided optimization of shape and texture perform? It would be nice to see the ablation study on this part to distinguish this method from LASSIE[39] and Hi-LASSIE[38].
2. How do you deal with the extra noise or overcorrection from the diffusion model?  The preprocessed image may look like an average animal in the species from the diffusion models.

**Limitations:**

The authors mentioned some limitations but not enough for me. Like the possible noise introduced from Stable Diffusion as mentioned above.

---

> ### Author Rebuttal · Authors · 2023-08-09
>
> ---
> **Contribution beyond LASSIE / Hi-LASSIE**
>
> Please see “Contribution beyond LASSIE / Hi-LASSIE” in the General Response above.
>
>
>
> ---
> **Unfaithful texture from Stable  Diffusion**
>
> Please see “Unfaithful texture from Stable Diffusion” in the General Response above.
>
>
> ---
> **“Figure 2 is not clear enough”**
>
> We thank the reviewer for the feedback and will update Figure 2 in the manuscript. We also provide a more detailed overview figure through this [anonymous link](https://www.dropbox.com/scl/fi/7abzk19c7cunubatnsp84/illustration.png?rlkey=hnlzefqxwvxf75hboktjhelw1&dl=0) and will add it to the supplemental document.
>
>
>
> ---
> **Ablation studies on distilling 3D reconstruction**
>
> Our ablation study of individual components is shown in Table 1 in the supplemental document, and we further report the detailed PCK gain (compared to Hi-LASSIE) in Table-T3 below. Note that the proposed DASS loss for 3D reconstruction brings 1.0-2.1% PCK gain from Hi-LASSIE without input preprocessing.
>
> **Table-T3**: PCK@0.05 on the E-LASSIE image sets (higher the better).
> | Method            | Input enhance. | $L_{dass}$ | Elephant    | Giraffe        | Kangaroo   | Penguin    | Tiger          | Zebra        |
> | :-----------        |  :-----------:      |  :-----------: | :-----------: | :-------:       | :-----------: | :---------:  | :-----:          | :-------:      |
> | Hi-LASSIE       |                         |                     |  37.6           | 54.3           | 31.9           | 41.7           | 57.4           | 60.1           |
> | ARTIC3D         |                         |  v                 |  38.8 (+1.2) | 56.1 (+1.8) | 34.0 (+2.1) | 42.7 (+1.0) | 58.5 (+1.1) | 61.9 (+1.8) |
> | ARTIC3D         |  v                     |                     |  39.0 (+1.4) | 57.3 (+3.0) | 34.6 (+2.7) | 43.4 (+1.7) | 58.5 (+1.1) | 62.4 (+2.3) |
> | ARTIC3D (full)  |  v                     |  v                 |  39.8 (+2.2) | 58.0 (+3.7) | 35.3 (+3.4) | 43.8 (+2.1) | 59.3 (+1.9) | 63.0 (+2.9) |

---

> > ### Comment · Reviewer_MsPL · 2023-08-14
> >
> > Thank the author for the detailed response.
> > Most of my concerns are resolved. I would raise my rating to 6.

---

> ### Author Response · Authors · 2023-08-14
> **Please let us know whether you have additional questions after reading our response**
>
> We appreciate your reviews and comments. We hope our responses address your concerns. Please let us know if you have further questions after reading our rebuttal.
>
> We hope to address all the potential issues during the discussion period.

---

### Author Rebuttal · Authors · 2023-08-09

# General Response

We thank the reviewers for the constructive feedback. We address the common concerns in the General Response and specific comments in the individual response to each reviewer.

---
### **Contribution beyond LASSIE / Hi-LASSIE**

While ARTIC3D deals with the same reconstruction task and leverages the skeleton/shape representation as in LASSIE and Hi-LASSIE, there are several key differences that allow ARTIC3D to handle occluded/truncated images and produce detailed texture in novel views. At a high level, LASSIE and Hi-LASSIE focus on **using geometry priors to learn detailed articulated shapes**, whereas ARTIC3D proposes to **incorporate generative 2D diffusion priors in a more challenging scenario with noisy images**. We emphasize that **combining such 3D geometry and 2D diffusion priors** is challenging, and that the proposed DASS module can effectively improve the results in all 3 stages (input preprocessing, 3D reconstruction, and animation).

As shown in Table 1 in the manuscript and Table-T1 below, Hi-LASSIE+ (Hi-LASSIE with SDS loss) marginally improves the PCK on E-LASSIE images by 0.1-1.2% by naively applying the common SDS loss, while ARTIC3D achieves a 1.9-3.7% PCK gain. Our CLIP similarity evaluation in Table 3 also shows consistently favorable textured reconstructions by ARTIC3D in different views. Moreover, our ablation study (supplemental Table 1 and Table-T2 below) demonstrates that the DASS loss for 3D reconstruction leads to 1.0-2.1% PCK gain from Hi-LASSIE without input preprocessing. We believe these PCK gains from ARTIC3D compared to prior works are considerable.

**Table-T1**: PCK@0.05 on the E-LASSIE image sets (higher the better).
| Method            | Elephant | Giraffe | Kangaroo | Penguin | Tiger | Zebra |
| :-----------          | :----------: | :-------: | :------------: | :---------: | :-----: | :------: |
| Hi-LASSIE       | 37.6           | 54.3            | 31.9            | 41.7           | 57.4            | 60.1           |
| Hi-LASSIE+     | 38.3 (+0.7) | 54.8 (+0.5) | 32.8 (+0.9) | 41.8 (+0.1) | 57.7 (+0.3) | 61.3 (+1.2) |
| ARTIC3D         | 39.8 (+2.2) | 58.0 (+3.7) | 35.3 (+3.4) | 43.8 (+2.1) | 59.3 (+1.9) | 63.0 (+2.9) |


**Table-T2**: PCK@0.05 on the E-LASSIE image sets (higher the better).
| Method            | Input enhance. | $L_{dass}$ | Elephant | Giraffe | Kangaroo | Penguin | Tiger | Zebra |
| :-----------          |  :-----------:        |  :-----------:   | :-----------: | :-------: | :-----------: | :---------: | :-----: | :-------: |
| Hi-LASSIE       |                         |                    |  37.6           | 54.3            | 31.9            | 41.7           | 57.4            | 60.1           |
| ARTIC3D         |                         |  v                |  38.8 (+1.2) | 56.1 (+1.8) | 34.0 (+2.1) | 42.7 (+1.0) | 58.5 (+1.1) | 61.9 (+1.8) |
| ARTIC3D (full) |  v                     |  v                |  39.8 (+2.2) | 58.0 (+3.7) | 35.3 (+3.4) | 43.8 (+2.1) | 59.3 (+1.9) | 63.0 (+2.9) |


---
### **Unfaithful texture from Stable  Diffusion**

Due to the highly ill-posed nature of our problem setting, there exists a tradeoff between realism and faithfulness to input images, especially for the unseen/occluded surface. For instance, whether a colored or black-and-white output is preferred given a greyscale/poorly illuminated image, or whether we preserve the noisy texture (small occlusions like dirt, water splash, or shadow caused by rough surface / objects) in the original image. Since the dense surface visibility is hard to obtain, we optimize the shape and texture that are slightly biased towards realism (detailed and clean texture that resembles the animal class) in this paper.

In our ablative analysis of the DASS module (Figure 3), we show that **we can control the realism-faithfulness tradeoff** by tuning the diffusion timestep $t$ and guidance weight $w_g$. Specifically, larger $t$ and $w_g$ allows DASS to hallucinate shape and texture that are not present in the original image. In addition, one can enforce a more faithful reconstruction by setting a higher weight of the texture reconstruction loss $\alpha_{text}$ (L223). We also show additional visual results on the tradeoff through this [anonymous link](https://www.dropbox.com/scl/fi/hshbqmf2gemq0ou32qa7f/faithfulness.png?rlkey=m6qt98hk5kuvuqoeoz96l2o3q&dl=0).

Although we acknowledge that the current tradeoff is not optimal and some example outputs are not faithful to the input images, our evaluation of image-to-image CLIP similarity in Table 3 shows that ARTIC3D outputs are still generally more faithful to input images across different views compared to the existing methods. We believe that ARTIC3D is a good first step in this novel scope, and automatically finding the best tradeoff for each animal class/instance forms an interesting future work.

---

### Decision · Program_Chairs · 2023-09-21

**Decision:**

Accept (poster)

**Comment:**

All the reviewers recognize the contributions and interest of the paper. The AC agrees and is happy to recommend acceptance.

The ablation results from the rebuttal seem essential to assess the contribution, it should be included in the final version of the paper.